

# Connectivity in Spanish metapopulation of Dupont's lark may be maintained by dispersal over medium-distance range and stepping stones

Alexander García-Antón[1], Vicente Garza[1] and Juan Traba[1,2]

[1] Terrestrial Ecology Group (TEG-UAM), Department of Ecology, Universidad Autónoma de Madrid, Madrid, Madrid, Spain
[2] Centro de Investigación en Biodiversidad y Cambio Global, Universidad Autónoma de Madrid, Madrid, Madrid, Spain

## ABSTRACT

**Background:** Dupont's Lark is an endangered bird, whose fragmented distribution in Europe is entirely restricted to Spain. This lark, suffering pronounced population decline and range contraction, inhabits steppes that have long been used for grazing sheep and are now threatened by rural abandonment and land use changes. Thus, for conservation of the lark, increasing knowledge about the connectivity of the Spanish metapopulation and identifying the most important connectivity nodes are crucial.

**Methods:** The study was carried out in Spain, using over 16,000 Dupont's Lark georeferenced observations. We used distance buffers to define populations and subpopulations, based on the available scientific information. We identified potential stepping stones using a MaxEnt probability of presence model. Connectivity was assessed using Conefor software, using the centroid of each subpopulation and stepping stone as nodes. Each node was assigned a quantitative attribute based on total habitat area, within-node habitat quality and internal fragmentation. We evaluated different connectivity scenarios by potential movement thresholds (5–20–100 km) and presence or absence of stepping stones in the network.

**Results:** Dupont's Lark Iberian metapopulation comprises 24 populations and 100 subpopulations, plus 294 potential stepping stones. Movement thresholds and stepping stones had a strong influence in the potential network connectivity. The most important nodes are located in the core of the metapopulation, which shows connectivity among subpopulations in the different indices and scenarios evaluated. Peripheral populations are more isolated and require stepping stones or medium (20 km) or long (100 km) potential movement thresholds to join the network.

**Discussion:** Metapopulation connectivity may be greater than expected, thanks to stepping stones and potential medium-distance movements. Connectivity is crucial for conservation and can be increased by preserving or improving adequate habitat in the most important nodes. Given the current species decline, steppe habitat should be urgently protected from land use changes and agriculture intensification, at least in the critical subpopulations and stepping stones. Long-term conservation of steppe lands and Dupont's Lark in Spain requires the recovery of traditional grazing and more research on juvenile dispersal. Meanwhile, the

Corresponding author
Juan Traba, juan.traba@uam.es

conservation of potentially critical stepping stones should be incorporated to management plans.

## INTRODUCTION

Connectivity of animal populations is of major importance for biodiversity conservation and plays a special role when managing threatened species (*Crooks & Sanjayan, 2006*; *Pascual-Hortal & Saura, 2006*). Both ecosystem functionality and population persistence depend on the degree of connectivity among the habitat patches, which is associated to the movement capacity of the focal species and to the landscape configuration (*Pascual-Hortal & Saura, 2007*). Patch isolation relies on factors such as size, distance to neighbours or the degree of permeability of the matrix. In general, small and isolated patches have a lower probability of occupancy than large and connected ones (*Levins, 1970*; *Hanski, 1999a*), depending on the ecology of the studied species: minimum patch size required (*Vögeli et al., 2011*; *Shake et al., 2012*), dispersal capacity (*Rolstad, 2008*) or matrix composition (*Ricketts et al., 2002*; *Vögeli et al., 2010*; *Watling et al., 2011*).

A population can occupy a group of isolated patches (fragments) if they are connected enough to permit dispersal and gene flow, thus forming a metapopulation (*Levins, 1969*; *Hanski, 1998*, *1999a*; *Hanski & Gaggiotti, 2004*). The probability of connection between two fragments depends on the dispersal ability of the species, which is linked to the distance between fragments, and characteristics of the matrix through which dispersal happens (*MacArthur & Wilson, 2001*; *Whittaker & Fernandez-Palacios, 2007*; *Losos & Ricklefs, 2009*). A patch can be completely isolated if the distance is too great for the dispersal ability of the species. In the traditional island biogeography theory, mainland areas are sources of individuals colonizing new areas, but in a metapopulation context immigration may occur from different habitat patches and populations (*Hanski, 1998*). Metapopulation dynamics will be determined in part by the quality, size, and distances between the fragments. From the connectivity perspective, the loss of a part of the metapopulation can have consequences for the rest, being more or less severe depending on the importance of the lost patch in the whole network (*Hanski, 1999a*, *1999b*).

Graph-based models are used to describe the movement-among-patch potential in a metapopulation comprising scattered habitat nuclei (patches or fragments) within an unsuitable matrix (*Pascual-Hortal & Saura, 2006*; *Bodin & Saura, 2010*; *Saura & Rubio, 2010*), and they offer quantitative information to identify critical patches for the maintenance of the functionality of the whole network (*Calabrese & Fagan, 2004*; *Visconti & Elkin, 2009*). A graph structure is based on two elements: nodes and links between them (*Saura & Torne, 2009*). Nodes represent suitable habitat patches occupied by the species or those acting as stepping stones (*Loehle, 2007*). Links are the connections between nodes, frequently estimated as the distance between them. Each node is also given a numerical

value that defines its quality within the network; usually, habitat area or other factor that describes focal species requirements (*Mazaris et al., 2013*; *Pereira, Saura & Jordán, 2017*).

Dupont's Lark (*Chersophilus duponti*; *Vieillot, 1824*, family Alaudidae) is a paradigmatic case for the study of connectivity, given the high degree of both natural and human-induced fragmentation of its habitat in Spain (*García-Antón et al., 2019*), and its strong specialization (*Suárez, 2010*). It is a small (~17.5 cm, ~38.5 g) bird that is extremely elusive, rare and, in Europe, only found in Spain, though it also occurs in northern Africa (*De Juana & Suárez, 2020*). The Spanish population is isolated from the African one and they are genetically and morphologically different (*García et al., 2008*; *García-Antón, Garza & Traba, 2018*). It is only found on mainly plain steppes (<15% slope), that in Spain have long been used by humans, especially for grazing sheep. Sheep tend to keep the vegetation low (<40 cm) and cause a large proportion of bare ground (*Garza & Suárez, 1990*; *Martín-Vivaldi et al., 1999*; *Garza et al., 2005*; *Garza et al., 2006*; *Nogués-Bravo & Agirre, 2006*; *Seoane et al., 2006*).

Isolation of populations is one of the major concerns for Dupont's larks in Spain (*Garza, Suárez & Tella, 2004*; *Íñigo et al., 2008*; *Méndez, Tella & Godoy, 2011*). Its fragmented distribution may function as a metapopulation (*Gómez-Catasús et al., 2018*; *García-Antón et al., 2019*; *Traba et al., 2019*), with different fragments or group of fragments having their own demographic parameters (*Pérez-Granados, López-Iborra & Garza, 2017*), with individual movements connecting them (*García-Antón, Garza & Traba, 2015*) and recolonization events of extinct patches (*Bota, Giralt & Guixé, 2016*). The sample bias toward adult males associated with the capture method (*Garza, Traba & Suárez, 2003*; *Suárez, 2010*) provides little information regarding other age and sex classes. Adults are sedentary (*Cramp & Simmons, 1980*; *Suárez et al., 2006*) and juveniles may disperse (*García-Antón, Garza & Traba, 2015*). The persistence of small and isolated subpopulations, however, suggests that medium to longer distance movements may often occur, from 4 to 300 km: ~5 km (*Pérez-Granados & López-Iborra, 2015*); 8 km (V. Garza, 2008–2019, unpublished data); 33 km (*García-Antón, Garza & Traba, 2015*). Some recent records reveal the existence of longer movements: 37 km (recolonization of Timoneda de Alfés, Lérida; *Bota, Giralt & Guixé, 2016*), 80 km (Salinas de Marchamalo, Murcia; *García & Requena, 2015*) and 99 km (Albufera de Valencia; *Dies et al., 2010*; *Balfagón & Carrion Piquer, 2021*), these being the minimum distance to the nearest occupied locality. Historic observations reveal even longer distance events: 127 km (Barcelona), 241 km (Trebujena-Sanlúcar, Cádiz), and up to 324 km (Marismas del Odiel, Huelva), among others (see Table S1).

Dupont's Lark occupies an area of around 1,000 km$^2$ in Spain, while another 965 km$^2$ of unoccupied habitat is available (*García-Antón et al., 2019*), which suggests that the metapopulation may be better connected than expected if this area is used as stepping stones. Recent studies indicate a generalized and pronounced decline in most Spanish subpopulations (*Gómez-Catasús et al., 2018*) and in their overall distribution (*García-Antón et al., 2019*). As fragmentation of the steppes continues, which is the main threat for Dupont's Lark (*Íñigo et al., 2008*), identification of critical patches for the maintenance of the metapopulation connectivity is required for the conservation of the species.

Here we address a detailed analysis of Dupont's Lark metapopulation connectivity in Spain, to provide a useful tool for the management and conservation of this threatened species. We hypothesize that the metapopulation must be better connected than expected, as connectivity and gene flow would explain the maintenance of the smallest and most isolated subpopulations, More specifically, we (i) update the cartography of populations and subpopulations of Dupont's Lark in Spain; (ii) identify both vulnerable and critical nodes from the connectivity point of view for the conservation of the metapopulation; (iii) assess the role of unoccupied but adequate regions in the metapopulation, testing the effect of different dispersal distance thresholds; (iv) evaluate the degree of isolation of each subpopulation; and (v) propose adequate conservation measures for the maintenance of the metapopulation.

## MATERIALS & METHODS

The ethics committee of Animal Experimentation of the Autonomous University of Madrid as an Organ Enabled by the Community of Madrid (Resolution 24th September 2013) for the evaluation of projects based on the provisions of Royal Decree 53/2013, 1st February, has provided full approval for this purely observational research (CEI 80-1468-A229).

### Species observations

We used the database of georeferenced observations of Dupont's Lark updated to 2017, including our unpublished data (TEG-UAM) and adding all available external records (*Traba et al., 2019*). We gathered a total of 17,755 Dupont's Lark locations corresponding to the temporal series of 1985–2017, both years included. We considered as recent those observations belonging to the post-2000 period ($n = 17,282$; 97%), when the II National Census was carried out (2004–2006; *Suárez, 2010*). This work allowed to standardize the field work using the territory mapping census method, which corrects the bias detected in previous works (*Garza, Traba & Suárez, 2003*; *Pérez-Granados & López-Iborra, 2013*). We considered that pre-2000 observations do not represent current species distribution patterns (see *García-Antón et al., 2019*), and so they were not included in this analysis.

Among the post-2000 locations, 14,203 came from our data (TEG-UAM), while the rest ($n = 3,079$) was provided by other administrations, research entities and individual ornithologists. We only used breeding season (February–July) observations. We excluded anomalous observations (clearly unoccupied or that only indicate moving animals). This resulted in 16,676 observations that we analyzed. These observations are aggregated in clusters, coinciding with the natural aggregation of habitat patches, though we have considered them as geographically independent for connectivity analyses.

### Species habitat

To build a map of Dupont's Lark habitat at a national scale we used CORINE land cover (CLC) inventory from the Copernicus European program, following the same method as in the distribution model (*García-Antón et al., 2019*). First, we intersected the 16,676

georeferenced observations with CLC 2006 layer (maintaining temporal correspondence with the period in which the majority of the observations belonged to, *i.e.*, II National Census, 2004–2006; *Suárez, 2010*). We selected the land use categories that accumulated 95% of the observations (see a description in Table S2), interpreting them as the habitat preferred by the species. Then, we extracted those categories from the most updated CLC available (2012) to get the current habitat map in Spain. To improve precision, we removed surfaces with slopes >15% (unsuitable habitat) and patches <20 ha (minimum threshold for species occupancy; *Suárez, 2010*). We used this map to estimate the habitat area within subpopulations and stepping stones (used as nodes in the connectivity model, see below). More details on the map building can be found in *García-Antón et al. (2019)*.

## Criteria for the definition of locality, subpopulation and population

We defined three sequentially hierarchical levels of actual occupancy by the species based on the map of 16,676 observations and distance thresholds published to date (*Laiolo, 2008*; *Suárez, 2010*; *Vögeli et al., 2010*; *Méndez et al., 2014*; *García-Antón, Garza & Traba, 2015*; *Bota, Giralt & Guixé, 2016*), as well as our unpublished data. Those were: locality, subpopulation and population.

We defined a locality as the area delimited by observations separated less than 1 km, distance that is known to be traveled by territorial males (*Suárez, 2010*; *Vögeli et al., 2010*). Data from capture-recapture of territorial adults indicate they are strongly sedentary, with regular movements <3 km (*Laiolo et al., 2007*; *Vögeli et al., 2008*; *Suárez, 2010*; *Vögeli et al., 2010*). Bioacoustic data suggest cultural similarity and adult males contact at a distance of 5 km (*Laiolo, 2008*), supported by the recovery of two marked adults at ~5 and 6 km in Rincón de Ademuz, Valencia (*Pérez-Granados & López-Iborra, 2015*). There is only one record of an adult out of this range, recaptured at 13 km from its capture location (V. Garza, 2008–2019, unpublished data). Thus, we established 5 km as the plausible threshold for resident movements. Therefore, a subpopulation was delimited by observations separated 5 km or less. Finally, a population was considered as the set of subpopulations separated by a maximum distance of 20 km, following a conservative criterion and accounting for the few available data on juvenile dispersal (up to 20 km in *Vögeli et al., 2010*, 33 km in *García-Antón, Garza & Traba, 2015*). This upper level represents those entities that, despite being connected sporadically would maintain a high genetic similarity due to individuals exchange (*Méndez, Tella & Godoy, 2011*; *Méndez et al., 2014*). We used a GIS software (*QGIS, 2021*) to build the correspondent buffers of 0.5, 2.5 and 10 km over the observations layer (Fig. 1).

## Definition of stepping stones

We also identified those areas that, despite being unoccupied by the species, could be potentially used and relevant in the connectivity process due to their high probability of presence, as shown in the distribution model (*García-Antón et al., 2019*). To do so, we used the 1 × 1 km cells considered to be of potential distribution (*n* = 5,575; those that accounted for a probability value higher than the mean of the 1,370 ones with confirmed presence, see *García-Antón et al., 2019*). After excluding cells that included buffers

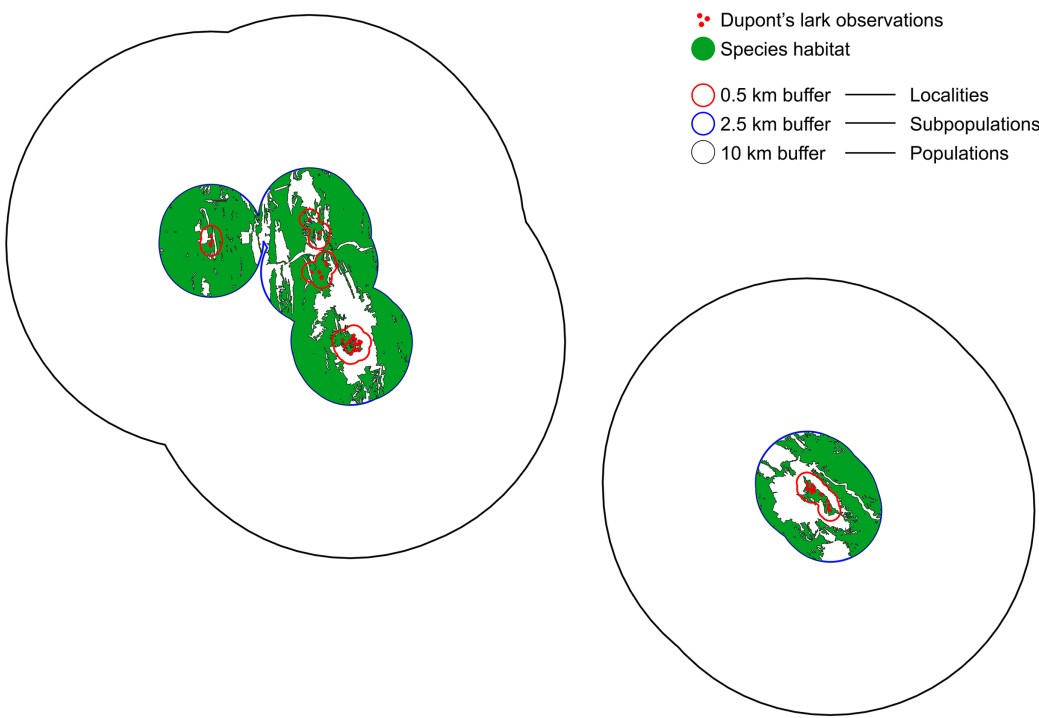

**Figure 1 Definition of localities, subpopulations and populations.** Localities are demarcated by a 0.5 km buffer (red), so that observations separated by a distance >1 km belong to different localities. Subpopulations are delimited by a buffer of 2.5 km (blue) and a distance of 5 km between observations. Finally, observations distanced >20 km belong to different populations (buffer of 10 km, black). Red dots indicate Dupont's lark observations and green polygons, the adequate habitat within subpopulations.

(that were included in the subpopulation layer), unoccupied potential habitat comprised 3,597 1 × 1 km cells. Adjacent cells were grouped into clusters, resulting in 902 independent entities. Following a conservative criterion, we removed those formed by a single 1 × 1 km cell, reducing it to 294 polygons. More details on the stepping stones building can be found in *García-Antón et al. (2019)*.

## Movement scenarios

The compilation of historic and recent Dupont's Lark observations out of the known breeding range (Table S1) reveals the existence of longer displacements than the thresholds defined previously, considered as rare events corresponding to sporadic long-distance movements. Taking into account all together, we defined three potential scenarios (see below): short (5 km); medium (20 km) and long (100 km) movements thresholds.

## Nodes and habitat attribute

We built the connectivity model at the subpopulation level, to obtain a more detailed result and considering that subpopulations, better than populations, constitute the metapopulation functional units, each with its own extinction risk and connectivity probability with the rest. This way, our network included one node located in the centroid of each subpopulation and stepping stone.

Each node was assigned a quantitative value that estimates its quality or importance in the network. We defined such attribute as Available Habitat Surface (AHS) and calculated it considering the surface of adequate habitat, its quality and its degree of fragmentation by intersecting the species habitat map (see above) with the subpopulation and stepping stone layer. Population size (number of territorial males) was not included in the AHS attribute as stepping stones account for no data on population size. Besides, we avoided bias in the result of our connectivity model toward historically occupied localities, regardless of their position in the actual metapopulation configuration Therefore, the AHS was defined as following:

$$AHS = HS \times HQ \times 1/NP$$

Where HS (habitat surface) is the total surface of adequate habitat within the subpopulation (or stepping stone), calculated as the sum of all habitat patches within each one; HQ is habitat quality, estimated as the mean value of probability of presence of the intersecting $1 \times 1$ km cells, as estimated in the MaxEnt model (*García-Antón et al., 2019*); and NP is the number of habitat patches within the subpopulation or stepping stone, as a measure of fragmentation. This way, each node obtained a value positively associated to its surface, quality and continuity of habitat.

To calculate the network links, we used the closest linear distance between borders of each pair of subpopulations and stepping stones, using a nearest neighbor algorithm in GIS software. We did not use distances between centroids because they do not reflect true distances that a bird would travel between patches, especially for larger patches.

## Connectivity model

We used software Conefor (*Saura & Torne, 2009*) to generate the connectivity model, which is widely used to analyse network structures (*Saura et al., 2011*; *Vergara et al., 2013*; *Grafius et al., 2017*). It builds the model in a two-step process: First, it calculates a connectivity index for the whole network (PC, probability of connection). It is based on node quality (AHS attribute), the distance between nodes, and dispersal capacity. Then, it removes each node independently and calculates the loss of PC due to that removal (dPC), obtaining an estimation of the contribution of each node to the global structure.

Conefor also allows the comparison between different general scenarios by means of the *equivalent connectivity index* (EC, see *Saura & Torne, 2009*), a modification of PC provided in the same units than the node attribute (see *Saura et al., 2011*; *Saura & Torne, 2009*). Prior to subsequent analyses, we compared scenarios resulting from the different movement thresholds considered (see above): short (5 km), medium (20 km) and long distance (100 km) and the presence or absence of stepping stones in the network (building the network with two different node maps, one including exclusively subpopulations and another one with the addition of all the stepping stones).

To evaluate the importance of each node for the network, dPC is fractioned into three more specific metrics: $dPC_{intra}$, $dPC_{flux}$ and $dPC_{connector}$ (*Pascual-Hortal & Saura, 2006*). The fraction $dPC_{intra}$ refers to the internal quality of the node (intra-patch connectivity), as it had been defined through the attribute considered (in this case, AHS). Thus, it is

independent of the distance to others nodes and its spatial position in the network. $dPC_{flux}$ is a value of inter-patch connectivity, giving information about the degree of flow that each node generates within the network; this index considers all the connections in which each node is either the origin or the destination points, as well as the quality of such connections (based on the AHS of the nodes involved). So, $dPC_{flux}$ depends on the spatial position of each node within the network, but also on the quality of those nodes it is connected to. Finally, $dPC_{connector}$ adds a second value of inter-patch connectivity, indicating the contribution of each node to the connectivity among the rest. This index provides information about the importance of each node for the maintenance of other nodes or group of nodes connectivity, that is, if it acts as a stepping stone whose absence would implicate that others increase their isolation or remain connected through a worse route (with a longer distance or passing through lower quality nodes). The total value of dPC is just the sum of these three fractions, so it gives a general value to each of the nodes in the network.

Finally, we calculated the matrix of probability of connection for each pair of nodes (subpopulations and stepping stones), what allows building connectivity maps for all different scenarios considered.

## RESULTS

### Populations, subpopulations and stepping stones

Based on the map of post-2000 observations and after the application of considered criteria we obtained 123 subpopulations, 23 of which are currently extinct, considering the most recent field data, updated to 2019. After removing them, we defined a present network of 100 subpopulations, 24 populations, plus the already mentioned 294 potential stepping stones (Fig. 2, Table S3, Data S1).

The metapopulation structure (Fig. 2) is formed by a core region comprising the largest population: Iberian Range—Ebro Valley (considered two independent populations to date, *Suárez, 2010*). Northwards, the metapopulation shows a myriad of small populations scattered through the Iberian Range (provinces of Soria, Zaragoza, Teruel, Navarra and Huesca), perhaps remnants of a historical more continued distribution. Further east and more isolated, the only Catalonian population: Alfés (Lérida province). Through the west (Zamora province) three small populations exist, with an apparent greater degree of isolation due to their distance with the core. Southwards, a group of 12 disperse populations and progressively more isolated from the core of the distribution are distributed along the provinces of Valencia, Cuenca, Toledo, Albacete, Murcia, Almería and Granada (Fig. 2, Data S1).

### Global connectivity under different scenarios

The EC index increased with the movement threshold and with the presence of stepping stones (Table S4). Because both movement threshold and stepping stones were important for connectivity, we include them both in all subsequent analyses.

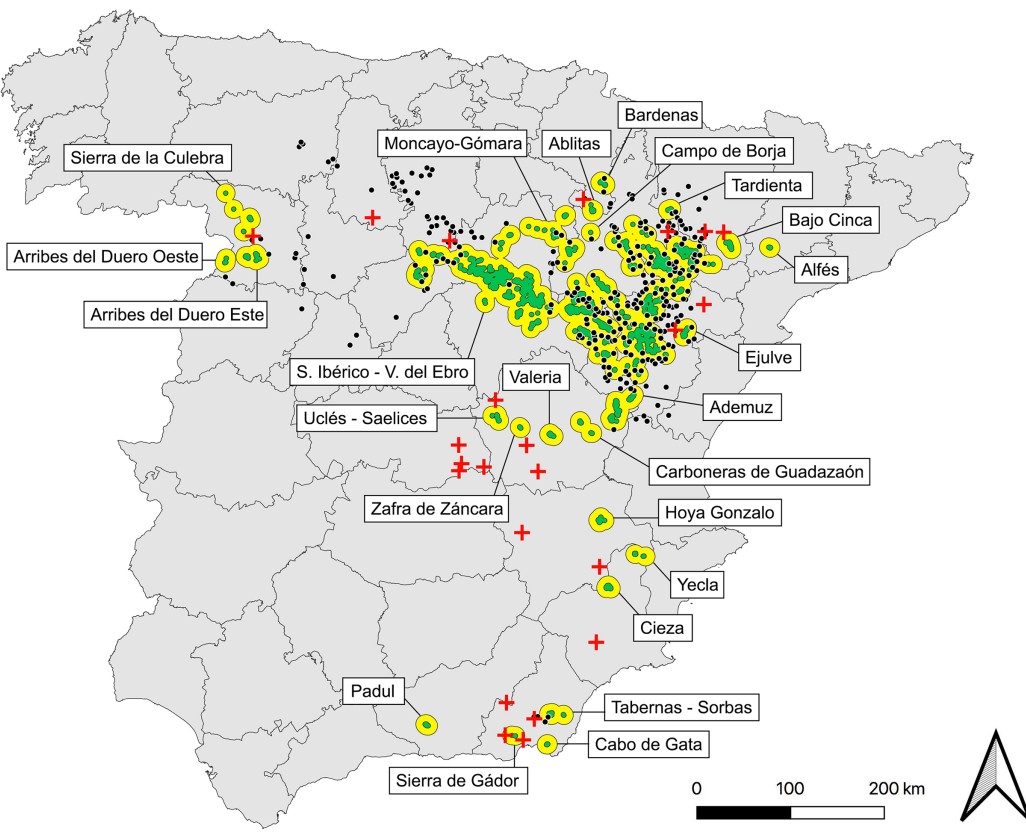

**Figure 2 Map of current populations, subpopulations and stepping stones of the Iberian metapopulation of Dupont's lark.** Black contours represent populations ($n = 24$), green polygons are subpopulations ($n = 100$) and black dots indicate stepping stones ($n = 294$). Red crosses represent the 23 subpopulations of recent extinction (post-2000). See detailed cartography in Data S1.

### Node classification by internal importance index (dPC$_{intra}$)

The subpopulations of *Monegros* (Z) and *Blancas* (TE) stand out with the highest dPC$_{intra}$ values (Table 1), meaning the best relation between habitat surface, quality and continuity (AHS attribute). The complete list (Data S2) shows two stepping stones in the first 20 positions: *Castronuño* (in Valladolid province, with the same dPC$_{intra}$ value than the 10th ranked subpopulation) and *Bardenas 2* (Navarra province).

### Node classification by importance for flow generation (dPC$_{flux}$)

The subpopulations of *Monegros* (Z) and *Blancas* (TE) were again the most important ones for this fraction, together with *Torralba de los Frailes* (TE), *Paramera de Molina* (GU) and *Gelsa* (Z) (Table 2). These subpopulations were those with more connections as starting or ending points, based on dPC$_{flux}$ values. No stepping stones were important when considering medium and long movement thresholds (20 or 100 km), but they appeared to be relevant in the scenario of short movements (5 km): *Monegrillo 2* (Z), *Alfajarín 1* (Z) and *Torralba de los Sisones* (TE) (top 10 ranking in Table 2, complete dataset is available in Data S2 and S3).

**Table 1 Summary of the 10 most important nodes for intra-patch connectivity (dPC$_{intra}$).**

| Name | Prov. | Without stepping stones (scenario 1, 2, 3) dPC$_{intra}$ | With stepping stones (scenario 4, 5, 6) dPC$_{intra}$ |
|---|---|---|---|
| Monegros | Zaragoza | 33.34 | 4.49 |
| Blancas | Teruel | 17.77 | 2.39 |
| Torralba de los Frailes | Teruel | 4.16 | 0.56 |
| Bardenas | Navarra | 2.64 | 0.36 |
| Lécera | Zaragoza | 2.10 | 0.28 |
| Pinilla del Campo | Soria | 1.49 | 0.20 |
| Campo Romanos | Zaragoza | 1.47 | 0.20 |
| Paramera de Molina | Guadalajara | 1.39 | 0.19 |
| Orihuela del Tremedal | Teruel | 1.25 | 0.17 |
| Gelsa | Zaragoza | 1.16 | 0.16 |

Note:
dPC$_{intra}$ makes reference to the internal importance of each node and it is independent on spatial position. Thus, the ranking is the same for the different movement thresholds. See the complete list in Data S2.

## Node classification by importance for connectivity maintenance (dPC$_{connector}$)

Three subpopulations, all included in the Iberian Range - Ebro Valley population, were the most important due to their function as connectivity nodes between others: *Paramera de Molina* (GU), *Layna* (SO) and *Altos de Barahona* (SO) (Table 3), followed by *Gelsa* (Z) and *Altiplano de Teruel* (TE), which were also present in all the scenarios. Four stepping stones were in top positions in the list: *Alba*, *Rubielos de la Cérida*, *Ojos Negros 1* and *Hoz de la Vieja*, all of them in Teruel province and within the Iberian Range - Ebro Valley population: (top 10 ranking in Table 3, complete dataset is available in Data S2 and S3).

## Node classification by general importance index (dPC)

Taking into account the sum of all previous fractions, *Monegros* (Z) and *Blancas* (TE) were highlighted as the most important subpopulations, followed by *Torralba de los Frailes* (TE) and *Paramera de Molina* (GU), all of them within the Iberian Range - Ebro Valley population (Table 4). When considering the presence of stepping stones, three important areas for the network connectivity were detected, also belonging to the same population: *Alba* (TE), *Rubielos de la Cérida* (TE) and *Cuerlas 1* (Z), which appear within the 10 most important nodes (Table 4). See Fig. 3 for a graphical view in an intermediate situation (scenario 5: 20 km movements and presence of stepping stones); the complete dataset is available in Data S2 and S3.

## Connectivity network

The degree of connectivity showed a strong variability under the different scenarios, highlighting the influence of potential movement thresholds and presence/absence of stepping stones in the metapopulation dynamics (Data S2 contains the complete matrix, with the probability of connection for each pair of nodes under each scenario).

**Table 2 Summary of the 10 most important nodes for flow generation in the network (dPC$_{flux}$).**

| Name | Prov. | dPC$_{flux}$ | Name | Prov. | dPC$_{flux}$ |
|---|---|---|---|---|---|
| **Scenario 1 (5 km mov. without SS)** | | | **Scenario 4 (5 km mov. with SS)** | | |
| Monegros | Zaragoza | 7.64 | Blancas | Teruel | 14.77 |
| Gelsa | Zaragoza | 7.06 | Monegros | Zaragoza | 14.03 |
| Torralba de los Frailes | Teruel | 5.82 | Torralba de los Frailes | Teruel | 7.64 |
| Paramera de Molina | Guadalajara | 5.33 | Paramera de Molina | Guadalajara | 6.11 |
| Blancas | Teruel | 3.52 | Gelsa | Zaragoza | 4.79 |
| Alforque | Zaragoza | 1.05 | Orihuela del Tremedal | Teruel | 4.23 |
| Pinilla del Campo | Soria | 1.02 | (E) Monegrillo 2 | Zaragoza | 3.62 |
| Milmarcos-Llumes | Guadalajara | 1.02 | Pozondón | Teruel | 3.11 |
| Pozalmuro | Soria | 0.89 | (E) Alfajarín 1 | Zaragoza | 2.69 |
| Cenegro | Soria | 0.82 | (E) Torralba de los Sisones | Teruel | 2.58 |
| **Scenario 2 (20 km mov. without SS)** | | | **Scenario 5 (20 km mov. with SS)** | | |
| Blancas | Teruel | 15.14 | Blancas | Teruel | 16.33 |
| Monegros | Zaragoza | 13.48 | Monegros | Zaragoza | 13.50 |
| Torralba de los Frailes | Teruel | 11.07 | Torralba de los Frailes | Teruel | 8.08 |
| Paramera de Molina | Guadalajara | 7.89 | Paramera de Molina | Guadalajara | 5.25 |
| Gelsa | Zaragoza | 7.81 | Orihuela del Tremedal | Teruel | 5.07 |
| Belchite | Zaragoza | 3.34 | Lécera | Zaragoza | 5.00 |
| La Torresaviñán | Guadalajara | 2.82 | Gelsa | Zaragoza | 3.74 |
| Lécera | Zaragoza | 2.81 | Belchite | Zaragoza | 3.48 |
| Cenegro | Soria | 2.61 | Pozondón | Teruel | 3.38 |
| Alforque | Zaragoza | 2.54 | Celadas Este | Teruel | 2.51 |
| **Scenario 3 (100 km mov. without SS)** | | | **Scenario 6 (100 km mov. with SS)** | | |
| Blancas | Teruel | 20.86 | Monegros | Zaragoza | 15.01 |
| Monegros | Zaragoza | 19.70 | Blancas | Teruel | 12.94 |
| Torralba de los Frailes | Teruel | 11.30 | Torralba de los Frailes | Teruel | 6.42 |
| Lécera | Zaragoza | 7.22 | Lécera | Zaragoza | 4.74 |
| Paramera de Molina | Guadalajara | 6.94 | Paramera de Molina | Guadalajara | 3.84 |
| Campo Romanos | Zaragoza | 5.49 | Orihuela del Tremedal | Teruel | 3.70 |
| Gelsa | Zaragoza | 5.35 | Belchite | Zaragoza | 3.44 |
| Orihuela del Tremedal | Teruel | 5.21 | Campo Romanos | Zaragoza | 3.43 |
| Belchite | Zaragoza | 5.09 | Gelsa | Zaragoza | 3.19 |
| La Torresaviñán | Guadalajara | 4.85 | La Torresaviñán | Guadalajara | 2.48 |

**Note:**
Stepping stones are indicated as '*SS*'. See the complete list in Data S2.

The most conservative situation (scenario 1: 5 km movements and absence of stepping stones) showed an extreme isolation, with connections among nearby subpopulations only in the metapopulation core (Fig. 4). Moreover, these connections seemed to be weak (0.001–20% probability), and lacking inter-population connections. In this situation, all the subpopulations outside of the Iberian Range—Ebro Valley population would be completely isolated. For this movement threshold, the presence of stepping stones would not be enough to connect the outermost subpopulations (scenario 4, Fig. 4).

**Table 3 Summary of the 10 most important nodes for connectivity maintenance (dPC$_{connector}$).**

| Name | Prov. | dPC$_{conn}$ | Name | Prov. | dPC$_{conn}$ |
|---|---|---|---|---|---|
| **Scenario 1 (5 km mov. without SS)** | | | **Scenario 4 (5 km mov. with SS)** | | |
| Paramera de Molina | Guadalajara | 2.38 | (SS) Alba | Teruel | 9.30 |
| Layna | Soria | 0.87 | Villar del Salz | Teruel | 6.89 |
| Altos de Barahona | Soria | 0.83 | (SS) Rubielos de la Cérida | Teruel | 6.70 |
| Gelsa | Zaragoza | 0.78 | Paramera de Molina | Guadalajara | 5.70 |
| Pozalmuro | Soria | 0.11 | (SS) Ojos Negros 1 | Teruel | 4.85 |
| Aldealpozo | Soria | 0.06 | (SS) Cuerlas 1 | Zaragoza | 4.68 |
| Cueva de la Hoz | Guadalajara | 0.04 | Blancas | Teruel | 3.84 |
| Altiplano de Teruel | Teruel | 0.02 | Pozondón | Teruel | 3.54 |
| Alforque | Zaragoza | 0.02 | (SS) Celadas | Teruel | 2.98 |
| Conquezuela | Soria | 0.01 | Monegros | Zaragoza | 2.37 |
| **Scenario 2 (20 km mov. without SS)** | | | **Scenario 5 (20 km mov. with SS)** | | |
| Paramera de Molina | Guadalajara | 6.65 | (SS) Alba | Teruel | 12.12 |
| Layna | Soria | 4.58 | Segura de los Baños | Teruel | 10.24 |
| Altos de Barahona | Soria | 3.57 | (SS) Rubielos de la Cérida | Teruel | 10.20 |
| Gelsa | Zaragoza | 2.60 | Villar del Salz | Teruel | 8.32 |
| Maranchón | Guadalajara | 1.55 | Altiplano de Teruel | Teruel | 8.26 |
| Villar del Salz | Teruel | 1.30 | Blancas | Teruel | 5.97 |
| Azaila | Teruel | 1.28 | (SS) Ojos Negros 1 | Teruel | 5.25 |
| Alforque | Zaragoza | 1.25 | (SS) Hoz de la Vieja | Teruel | 5.08 |
| Blancas | Teruel | 1.03 | (SS) Moneva | Zaragoza | 4.75 |
| Altiplano de Teruel | Teruel | 0.91 | Paramera de Molina | Guadalajara | 4.41 |
| **Scenario 3 (100 km mov. without SS)** | | | **Scenario 6 (100 km mov. with SS)** | | |
| Layna | Soria | 8.28 | Segura de los Baños | Teruel | 7.91 |
| Segura de los Baños | Teruel | 8.22 | Layna | Soria | 4.90 |
| Paramera de Molina | Guadalajara | 7.89 | (SS) Rubielos de la Cérida | Teruel | 4.09 |
| Altos de Barahona | Soria | 7.08 | (SS) Alba | Teruel | 4.07 |
| Altiplano de Teruel | Teruel | 3.78 | Altiplano de Teruel | Teruel | 4.03 |
| Blancas | Teruel | 3.60 | Altos de Barahona | Soria | 3.97 |
| Maranchón | Guadalajara | 2.86 | Paramera de Molina | Guadalajara | 3.80 |
| Azaila | Teruel | 2.52 | (SS) Pinilla Trasmonte | Burgos | 3.48 |
| Lécera | Zaragoza | 2.50 | (SS) Hoz de la Vieja | Teruel | 3.06 |
| Gelsa | Zaragoza | 2.16 | Villar del Salz | Teruel | 2.76 |

**Note:**
Stepping stones are indicated as '*SS*'. See the complete list in Data S2.

For potential movements up to 20 km (scenario 2, Fig. 4), the situation changed notably. Despite the connections among nearby subpopulations continued being of low-to-medium probability, inter-subpopulation connectivity occurred within the Iberian Range—Ebro Valley population and within the western populations. With the presence of stepping stones (scenario 5, Fig. 4), high probability connections (over 80%) were frequent in near all the subpopulations within and north to the Iberian Range—Ebro Valley population. The most western populations increased their inter-subpopulation connectivity but

**Table 4 Summary of the 10 most important nodes for the connectivity based on the global index dPC.**

| Name | Prov. | dPC | Name | Prov. | dPC |
|---|---|---|---|---|---|
| **Scenario 1 (5 km mov. without SS)** | | | **Scenario 4 (5 km mov. with SS)** | | |
| Monegros | Zaragoza | 40.99 | Monegros | Zaragoza | 28.96 |
| Blancas | Teruel | 21.29 | Blancas | Teruel | 25.30 |
| Torralba de los Frailes | Teruel | 9.98 | Paramera de Molina | Guadalajara | 12.33 |
| Paramera de Molina | Guadalajara | 9.10 | (E) Alba | Teruel | 11.22 |
| Gelsa | Zaragoza | 8.99 | Torralba de los Frailes | Teruel | 9.49 |
| Bardenas | NA | 2.64 | Villar del Salz | Teruel | 7.96 |
| Pinilla del Campo | Soria | 2.51 | Gelsa | Zaragoza | 7.37 |
| Lécera | Zaragoza | 2.27 | (E) Rubielos de la Cérida | Teruel | 7.35 |
| Orihuela del Tremedal | Teruel | 1.90 | Pozondón | Teruel | 6.83 |
| La Torresaviñán | Guadalajara | 1.77 | (E) Cuerlas 1 | Zaragoza | 6.43 |
| **Scenario 2 (20 km mov. without SS)** | | | **Scenario 5 (20 km mov. with SS)** | | |
| Monegros | Zaragoza | 33.79 | Blancas | Teruel | 24.70 |
| Blancas | Teruel | 26.85 | Monegros | Zaragoza | 20.46 |
| Paramera de Molina | Guadalajara | 15.37 | (E) Alba | Teruel | 14.04 |
| Torralba de los Frailes | Teruel | 13.62 | Segura de los Baños | Teruel | 11.67 |
| Gelsa | Zaragoza | 11.10 | (E) Rubielos de la Cérida | Teruel | 10.93 |
| Layna | Soria | 5.77 | Altiplano de Teruel | Teruel | 10.31 |
| Altos de Barahona | Soria | 4.89 | Paramera de Molina | Guadalajara | 9.84 |
| Belchite | Zaragoza | 4.73 | Villar del Salz | Teruel | 9.40 |
| Lécera | Zaragoza | 4.42 | Torralba de los Frailes | Teruel | 8.76 |
| Alforque | Zaragoza | 3.89 | Belchite | Zaragoza | 6.74 |
| **Scenario 3 (100 km mov. without SS)** | | | **Scenario 6 (100 km mov. with SS)** | | |
| Blancas | Teruel | 27.40 | Monegros | Zaragoza | 17.19 |
| Monegros | Zaragoza | 25.50 | Blancas | Teruel | 16.28 |
| Paramera de Molina | Guadalajara | 15.06 | Segura de los Baños | Teruel | 9.14 |
| Torralba de los Frailes | Teruel | 12.05 | Paramera de Molina | Guadalajara | 7.69 |
| Segura de los Baños | Teruel | 10.18 | Torralba de los Frailes | Teruel | 6.60 |
| Lécera | Zaragoza | 10.06 | Belchite | Zaragoza | 6.20 |
| Layna | Soria | 9.67 | Layna | Soria | 5.59 |
| Altos de Barahona | Soria | 8.67 | Altiplano de Teruel | Teruel | 5.46 |
| Gelsa | Zaragoza | 7.70 | (E) Alba | Teruel | 5.36 |
| Belchite | Zaragoza | 7.08 | Lécera | Zaragoza | 5.17 |

**Note:**
Stepping stones are indicated as '*SS*'. See the complete list in Data S2.

remained unconnected with the metapopulation core. The situation of the southern part of the distribution remained dramatically unconnected, even considering the presence of stepping stones (scenario 5, Fig. 4).

Only with potential movements up to 100 km (scenarios 3 and 6, Fig. 4), Dupont's Lark Iberian metapopulation would be completely connected, although even for this distance

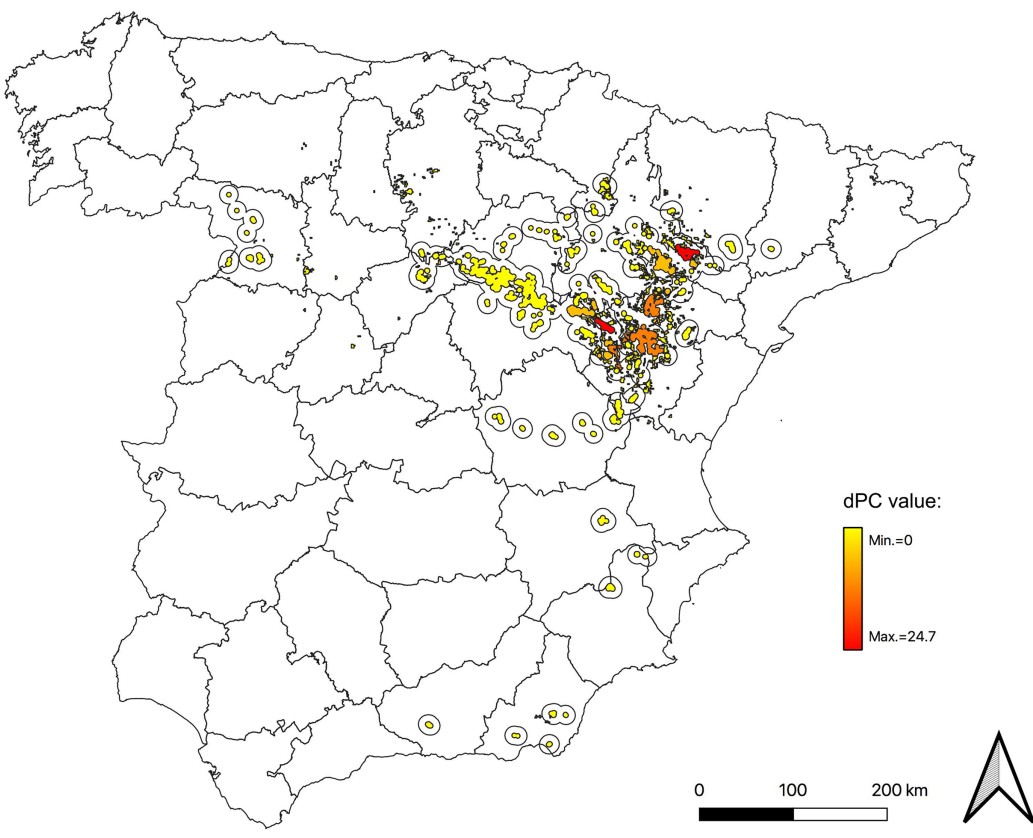

**Figure 3 Map of node importances in the Iberian metapopulation of Dupont's lark.** Nodes classified by general importance index (dPC). The core of the distribution, focused in the Iberian Range—Ebro Valley population, gathers the most important nodes. Here we show scenario 5 (movements of 20 km and presence of stepping stones). Maps for all possible scenarios are included in Data S2.

threshold, the absence of stepping stones (scenario 3) would result in weak connections of the western and southern subpopulations with the metapopulation core.

## DISCUSSION

The criteria applied in this work for the definition of localities (habitat patches separated by less than 1 km), subpopulations (group of localities separated 5 km or less) and populations (set of subpopulations separated by a maximum distance of 20 km) led to a Dupont's Lark metapopulation in Spain formed by 24 populations and 100 subpopulations. This metapopulation is probably dynamic and therefore should be periodically updated with continuous monitoring. Twenty-three additional subpopulations became extinct in the last two decades and should be regularly monitored to verify possible recolonizations.

Dupont's Lark seems not to fit a classic Levins model of colonization-extinction balance. On the contrary, extinctions seem to be permanent, in a source-sink pattern that reveals a contraction process from the peripheral subpopulations to the core of the distribution. Many adequate habitat patches ($n$ = 294) are spread out along the distribution range, although they are heterogeneously distributed. The distant western populations might be

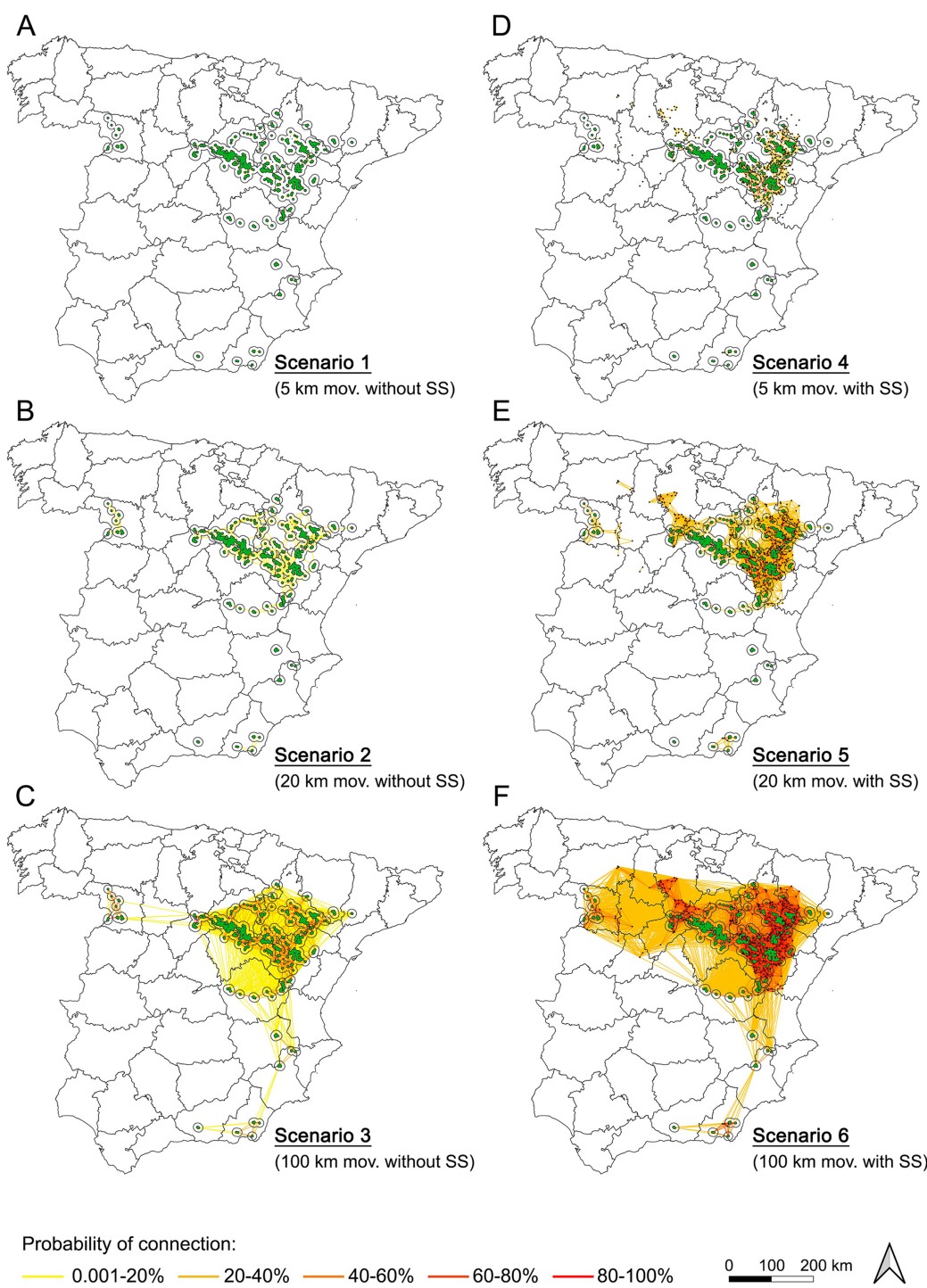

**Figure 4 Probability of connection of Dupont's lark metapopulation under the different scenarios evaluated (A–F).** Effect of the distance (movements of 5, 20 and 100 km) and the presence/absence of stepping stones in the probability of connection among Dupont's lark subpopulations. See Data S2 for the complete matrix of probability of connection for node pairs.

better connected than expected due to stepping stones. The southern range, however, is critically isolated and where the majority of recent subpopulation extinctions occurred.

Through this study we indicate stepping stones and subpopulations that are critical for connectivity. This information can be used by management to avoid increased habitat loss. Conservation measures should include steppe land habitat protection: avoiding infrastructures installation and land use changes, restoring habitat structure with active management and introducing traditional grazing to allow long-term conservation.

While dispersal mechanisms are poorly understood, our results suggest that movement over medium distances (~20 km) with stepping stones together help explain why small and isolated populations persist, rather than become extinct as previously predicted (*Traba, De la Morena EL & Garza, 2011*; *Suárez & Carriles, 2010*; *Laiolo et al., 2008*). This is supported by a recent study in Rincón de Ademuz (Valencia, eastern Spain), which obtained only one recovery out of 26 juvenile individuals marked, suggesting that juveniles either leave their natal site and disperse, or their survival rate is very low (*Pérez-Granados, Sáez-Gómez & López-Iborra, 2021*).

## Populations, subpopulations and stepping stones

Our results indicate that a large, single population comprises what was previously thought to be the two main populations (Iberian Range, Ebro Valley). The map of subpopulations indicates continuity in the core of the metapopulation and has a strong degree of fragmentation and isolation southwards and in the western range. Our results support the high vulnerability of the peripheral subpopulations, as showed previously in the Ebro Valley (*Vögeli et al., 2010*) and in genetic analysis (*Méndez, Tella & Godoy, 2011*), which are more prone to extinction (*Méndez et al., 2014*; *Gómez-Catasús et al., 2018*).

While there are many potential stepping-stones (almost 300), they are still likely to be important for metapopulation dynamics. Most of them are in the easternmost distribution (Teruel and Zaragoza provinces). The lack of stepping stones along the southern range helps to explain the dramatic trends of the southernmost subpopulations (*Gómez-Catasús et al., 2018*). The apparently strict isolation of the western range (Zamora province) may be surprisingly well-connected due to the many stepping-stones (Fig. 2). Most of the areas along the metapopulation with apparent optimal habitat but absence of the species (*García-Antón et al., 2019*) are considered as stepping stones in this work, and they might play a role in the species movements. Whether these areas correspond to empty patches in a classic colonization-extinction balance (*Levins, 1969*) remains unknown. However, population turnover in Dupont's Lark seems to be extremely rare in both metapopulation (García-Antón, Garza & Traba, 2021, under review) and local scales (*Gómez-Catasús, Garza & Traba, 2018*). To our knowledge, just one known subpopulation has been recolonized after going extinct (*Bota, Giralt & Guixé, 2016*). Intensive field work in the Iberian Range along the study period has recorded one single habitat patch (within a known locality) reoccupied (own data). Rather than a classical Levins model, Dupont's Lark metapopulation could adopt a source-sink structure (*Hanski, 1998*, *1999a*). The smaller and more isolated subpopulations would be in a higher risk of extinction due to its lower connectivity with the core of the distribution, besides other risks associated to

its lower size. More than 50% of the Iberian subpopulations have less than 5 individuals (*Traba et al., 2019*), which from a genetic and demographic point of view suggests low medium-term viability, if there is no connection with other subpopulations (*Méndez, Tella & Godoy, 2011*; *Méndez et al., 2014*).

Those subpopulations that went extinct during the post-2000 period ($n$ = 23, which means 19% of the extant subpopulations at the beginning of the century) could correspond to stochastic factors or to changes in habitat quality (*Hanski, 1999a*). In the first case, such patches would be immediately available for recolonization, as the one recorded by *Bota, Giralt & Guixé (2016)* in Alfés (Lérida) in 2015. In the latter, that subpopulation would be unavailable for recolonization until habitat was restored. There are two main factors promoting habitat loss in the case of Dupont's Lark. First, the abandonment of extensive grazing leads to plant succession and transformation of the steppe land habitat (*Peco et al., 2012*; *Íñigo et al., 2008*; *Gómez-Catasús et al., 2019*), in addition to decreasing habitat quality due to food (arthropod) availability linked to sheep deposition (*Gómez-Catasús et al., 2019*; *Reverter et al., 2019*). Second, direct habitat destruction due to land use changes, mainly wind farms (*Gómez-Catasús, Garza & Traba, 2018*) and ploughing (*Garza, Suárez & Tella, 2004*; *Íñigo et al., 2008*), and new habitat changes expected to appear in the near future (wind farms and solar photovoltaic installations; *Serrano et al., 2020*).

Therefore, two key elements are crucial for Dupont's Lark conservation: habitat maintenance in the areas inhabited by the species (or those considered important for the connectivity network) and the promotion of active management to guarantee long-term habitat persistence. Recent initiatives in this direction have been positive (LIFE Ricotí in Soria, local projects in Valencia region; see a revision in *Traba et al. (2019)*), and will be a useful tool for key areas (such as critically isolated subpopulations or important stepping stones). Anyway, long-term effective measures for habitat and species conservation should include the promotion of traditional sheep grazing, in order to avoid dramatic plant structure changes and maintain habitat functionality. These measures should be considered, at least, in the most critical connectivity nodes.

Regarding the extinct subpopulations, only 7 of 23 have become stepping stones based on our habitat-suitability criteria. This result suggests that low habitat quality (*i.e.* low food availability, changes in vegetation structure) in those areas may have contributed to the local extinction of the species, in addition to isolation. Indeed, 14 out of these 23 extinct subpopulations are located in the southern range (Fig. 2), where isolation is more accused, following a centripetal contraction process from the periphery to the metapopulation core (García-Antón, Garza & Traba, 2021, under review).

In Datas S1, S2 and S3 we offer detailed data and updated cartography of the metapopulation that can constitute a useful guide for the different regional administrations, which have legal obligations for the conservation of Dupont's Lark in Spain. Management coordination and common guidelines are of vital importance in the case of Dupont's Lark, as several regional administrations are affected by its distribution and share populations or subpopulations.

## Global connectivity under different scenarios

Despite the apparent strong fragmentation and high degree of isolation of Dupont's Lark metapopulation, our results suggest two elements that seem to be relevant for the connectivity of the whole network. These factors may contribute to explain the prevalence of the smallest and most isolated subpopulations, which were expected to be extinct based on the population viability models (*Laiolo et al., 2008*; *Suárez, 2010*), genetic structure (*Méndez, Tella & Godoy, 2011*; *Méndez et al., 2014*), and data on the general situation of the species (*Suárez, 2010*; *Traba et al., 2019*). First, the large area of vacant adequate habitat (*García-Antón et al., 2019*), that should be interpreted as a network of stepping stones unnoticed to date. The size of this stepping stone network approximately equals the size of the occupied range of Dupont's Lark (around 1,000 km$^2$; *García-Antón et al., 2019*). The Equivalent Connectivity index (EC) comparison (Table S4) showed the lowest value of EC for scenario 1 (5 km movement threshold without stepping stones), while EC for scenario 6 (100 km movement threshold with stepping stones) had the highest value. For each scenario, EC was always higher when adding stepping stones than increasing potential movements to the next threshold. Therefore, the role of these unoccupied potential areas seems crucial for the functionality of the network and could have even a stronger influence than the movement capacity of the species (Table S4). In other words, even if we consider Dupont's Lark as a strongly sedentary species with sporadic medium-distance movements, the metapopulation could be connected thanks to the presence of stepping stones. The relative low values of stepping stones in $dPC_{intra}$ (Table 1) but higher ones in $dPC_{flux}$ and $dPC_{connector}$ (Tables 2 and 3) suggest that these patches may have lower habitat quality than occupied subpopulations (based on the AHS attribute), thus being unsuitable for occupancy, but maintaining a high relevance for the metapopulation connectivity.

On the other hand, results of the simulation of different movement thresholds (Fig. 4) suggest that 2-5 km maximum dispersal distance assumed previously (*Laiolo et al., 2007*; *Vögeli et al., 2008*; *Vögeli et al., 2010*; *Suárez, 2010*) could have undervalued actual dispersal ability of the species. Recent records of longer movements, that could correspond to juvenile dispersal (García-Antón, 2015), recolonization (*Bota, Giralt & Guixé, 2016*) or sporadic long-distance movements (*García & Requena, 2015*, *Dies et al., 2010*, *Balfagón & Carrion Piquer, 2021*), as well as historical records summarized in Table S1, point to medium to large distance events that could be contributing to slow down local extinction as fast as predicted by the viability models (*Laiolo et al., 2007*; *Suárez, 2010*).

## Node importance and AHS attribute

$dPC_{intra}$, $dPC_{flux}$ and dPC indicated the same most important nodes: *Monegros* (Z), *Blancas* (TE), *Torralba de los Frailes* (TE) and *Paramera de Molina* (GU), all of them located in the Iberian Range—Ebro Valley population. The conservation of these top ranked subpopulations is imperative to ensure the conservation of the metapopulation, as it is also crucial to focus on the third fraction of dPC ($dPC_{connector}$). In the case of Dupont's Lark, in which isolation may constitute a critical factor for the species conservation, the loss of those subpopulations with a higher value in $dPC_{connector}$ could implicate the

subsequent extinction of other subpopulations or groups of subpopulations, so they should be considered of highest priority. Several nodes of the Iberian Range close to the geographical centroid of the metapopulation are included in this set, mainly *Layna* (SO), *Paramera de Molina* (GU) and *Altos de Barahona* (SO), as well several stepping stones that are also among the top ranked nodes: *Alba*, *Rubielos de la Cérida*, *Ojos Negros 1* and *Hoz de la Vieja*, among others (Table 3).

Finally, the particular case of the military National Training Centre of *San Gregorio*, a few km North of Zaragoza city, must be considered. This area holds around 34,000 ha of mostly continuous steppe habitat and due to its huge extension it might certainly constitute one of the most important nodes of the connectivity network. In determining stepping-stones, we identified habitat (stepping stones Zaragoza 1, 3, 4, 5, 6) that is potentially important, and should be treated as such by the regional administration of Aragón.

Data S2 includes the complete lists of node importance by province in all the scenarios considered and should constitute a useful management tool. Each regional administration should consider the most important nodes within its territory, either subpopulations or stepping stones, of high priority and concern. These areas should be included in national and/or regional species conservation plans, as their protection and management seem to be crucial for the maintenance of the species at a national scale, and coordinated measures between neighbour administrations are needed. Stepping stones require special attention, as they are relevant for their spatial and habitat features, but not for the presence of the species, which may difficult the application of conservation measures.

## Connectivity network

In the most restrictive scenario (movements of 5 km and absence of stepping stones), the subpopulations were almost totally isolated, except for the low probability connections within the Iberian Range—Ebro Valley. Assuming a medium movement threshold of 20 km, a significant increase of connections appears within the central distribution, though their probability continued being low. Thus, the uttermost western populations seem to be isolated and their persistence depend on the presence of stepping stones. The most unfavorable situation is in the southern subpopulations, which remain completely isolated without movements of 100 km.

The strong population decline of the species (*Gómez-Catasús et al., 2018*), its current and future distribution (*García-Antón et al., 2019*), and the genetic analyses (*Méndez, Tella & Godoy, 2011*; *Méndez et al., 2014*) indicate important degree of isolation. But, at the same time, small and isolated peripheral subpopulations persist. Therefore, we suggest that some combination of our scenarios is most likely. Based on movements of the different age classes, and with the little information on juvenile capture-recapture, we suggest that adult movements less than 1 km are very likely (high probability), and so intra and inter-sexual communication at this distance must be common. Adult movements between 1 and 5 km could be mid-to-low probability events; those between 5 and 20 km, of low probability; and those over 20 km must be considered highly improbable events. Juveniles

are presumable the dispersive fraction of the population, as it is widespread in other bird species (*Weise & Meyer, 1979*; *Greenwood & Harvey, 1982*; *Ferrer, 1993*; *Cooper, Daniels & Walters, 2008*; *Whitfield et al., 2009*). Juveniles tend to disperse (as recently suggested for Dupont's Lark, *Pérez-Granados, Sáez-Gómez & López-Iborra, 2021*), moving long distance across non-habitat areas and to settle new populations with few initial individuals (*Harrison, 1989*). In the case of the Dupont's Lark, juvenile movements of 5 km are very likely; those comprising 5–20 km of high probability; 20–100 km of low probability; over 100 km of very low probability. This last distance would be rare events of sporadic long-distance movements (Table S1).

The importance of stepping stones facilitating movements between habitat fragments has been reported in different ecosystems and species. *Uezu, Beyer & Metzger (2008)* showed in the bird community of the Brazilian Atlantic forest that the efficiency of stepping stones is species-dependent and related to matrix resistance. *Baum et al. (2004)* also highlighted the importance of the surrounding matrix for the effectiveness of stepping stones in plants. *Saura, Bodin & Fortin (2014)* found that the loss of stepping stones can cause a sharp decline in the potential movement distance in bird species, which are not compensated for other factors (*e.g.*, source population size). Stepping stones could also have some negative effects, as *Kramer-Schadt et al. (2011)* found in a mammal species, with a trade-off related to stepping stone size and location.

The situation of Dupont's Lark shows dramatic declines and ongoing habitat fragmentation and contraction (*Gómez-Catasús et al., 2018*; *García-Antón et al., 2019*), urgently suggests that immediate management of the species and habitat are necessary. In the current context of land intensification and rural abandonment, Dupont's Lark habitat has a finite lifetime. As smaller patches disappear, the larger ones, which presently hold the majority of the population, will become more vulnerable due to the loss of linked habitat and the decrease of connectivity. Besides, several aspects of this species remain partially unknown and are crucial for its conservation, as dispersal mechanisms, reproductive biology or genetics, which are needed for a detailed evaluation of the connectivity and population viability of Dupont's Lark.

## CONCLUSIONS

Conservation and management of the Dupont's Lark in Spain is urgent, and here we list the most important areas to carry that out. Habitat loss and fragmentation must be urgently stopped in Dupont's Lark subpopulations and stepping stones. This is mainly being produced by ploughing, windfarms and afforestation. Additionally, the increase of habitat quality both in short (restoration measures) and long terms (extensive grazing) is desirable for the species conservation. Isolation of the southern range is extreme and, due to the recent subpopulation extinctions, we speculate a near-future distribution restricted to the current metapopulation core. Research on movements, especially on breeding dispersal, would help clarifying movement patterns in the metapopulation and establishing ecological corridors to increase connectivity.

## ACKNOWLEDGEMENTS

We wish to acknowledge to Francisco *Quico* Suárez, who led the Dupont's Lark research group until his early death in 2010. Many people collected data in the field; we especially wish to thank: A. Agirre, R. Aymí, M. Calero, E. Carriles, J. T. García, I. Hervás, J. H. Justribó, E. G. de la Morena, J. J. Oñate and J. Viñuela. All birds were captured and processed following the Wild Birds Ringing Manual and under the correspondent official licenses. This is a contribution to the Excellence Network Remedinal 3CM (S2013/MAE-2719), supported by Comunidad de Madrid.

### Funding

The data used in this article are owned by or came from different projects headed by the Terrestrial Ecology Group (Universidad Autónoma de Madrid) and Juan Traba: in particular the II National Census (2004–2006), granted by the Spanish Ministry of Environment; the Dupont's Lark Monitoring Program in Medinaceli Region, funded by Fundación Patrimonio Natural de Castilla y León, and the projects "Criteria for the management and conservation of the Spanish population of Dupont's Lark", supported by Fundación Biodiversidad, of the Ministry of Agriculture, Food and Environment; LIFE Ricotí (LIFE15-NAT-ES-000802), supported by the European Commission; and BBVA-Dron Ricotí, funded by the BBVA Foundation. The funders had no role in study design, data collection and analysis, decision to publish, or preparation of the manuscript.

### Grant Disclosures

The following grant information was disclosed by the authors:
Terrestrial Ecology Group (Universidad Autónoma de Madrid).
II National Census (2004–2006).
Spanish Ministry of Environment.
Dupont's Lark Monitoring Program in Medinaceli Region.
Fundación Patrimonio Natural de Castilla y León.
Fundación Biodiversidad, of the Ministry of Agriculture, Food and Environment.
LIFE Ricotí: LIFE15-NAT-ES-000802.
European Commission.
BBVA-Dron Ricotí.
BBVA Foundation.

### Competing Interests

The authors declare that they have no competing interests.

### Author Contributions

- Alexander García-Antón performed the experiments, analyzed the data, prepared figures and/or tables, authored or reviewed drafts of the paper, and approved the final draft.

- Vicente Garza performed the experiments, authored or reviewed drafts of the paper, prepared and curated data, and approved the final draft.
- Juan Traba conceived and designed the experiments, performed the experiments, analyzed the data, authored or reviewed drafts of the paper, and approved the final draft.

## Animal Ethics

The following information was supplied relating to ethical approvals (*i.e.*, approving body and any reference numbers):

The ethics committee of Animal Experimentation of the Autonomous University of Madrid as an Organ Enabled by the Community of Madrid (Resolution 24th September 2013) for the evaluation of projects based on the provisions of Royal Decree 53/2013, 1st February, has provided full approval for this purely observational research (CEI 80-1468-A229).

## Data Availability

The raw data was available for review and available upon request since the georeferenced observations for an endangered species are sensitive.

## Supplemental Information

Supplemental information for this article can be found online at http://dx.doi.org/10.7717/peerj.11925#supplemental-information.

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
