# Peer review of "Connectivity in Spanish metapopulation of Dupont’s lark may be maintained by dispersal over medium-distance range and stepping stones"

_PeerJ, doi:10.7717/peerj.11925_

## Round 0.1 · original submission · Major Revisions

· Academic Editor

Major Revisions

While the two reviewers had very different perspectives on the manuscript, I agree that the changes are large. As neither of the reviewers are native English speakers, I took it upon myself to insert quite a few suggestions in the annotated manuscript that I also send. As the reviewers noted, some terminology was ambiguous in its use, and so please be careful to use the correct word in the correct context (e.g., dispersal, dispersion, etc.).

Rather than repeat the comments of the two reviewers, with which I agree, I mostly added comments about the writing style and some logic and consistency, that is also very relevant for clear communication. Rather than repeat my comments here, you should read them in context in the annotated manuscript.

The second reviewer pointed out some logical inconsistencies with the numbers and the analyses - for example, the important fragment that only had 3 adult males, as compared to a less important fragment with many - and so I want you to pay particular attention to those comments.

Please provide a detailed rebuttal for both reviewers and my comments.

Reviewer 1 ·

Basic reporting

No comment.

Experimental design

No comment

Validity of the findings

No comment

Additional comments

This manuscript analysis the connectivity of the endangered Dupont’s lark metapopulation in its fragmented habitat in Spain, comparing different dispersal distances and the effect of accounting for unoccupied potential habitat patches as stepping stones. I think the topic is very interesting and the manuscript is nicely written. The methods are appropriate and the results highlight the key role that stepping stones habitat patches may play in the population viability of the species.

I only have some minor comments about the manuscript:

Abstract:
Although the main text is very nice to read, the abstract is somehow difficult to follow. I suggest rewriting it including some general background for the study in addition to the objectives; in methods, more specific information that includes how connectivity was assessed and the different dispersal distances that were used; same for the results.

Introduction:
- Line 50. …isolation from the rest…
- Line 84. …habitat area?...
- Lines 93-97. This is a long sentence that is a bit difficult to follow. I suggest to rephrase it.

Materials and methods:
- Subsection “Species habitat”: This section is a bit unclear to me. If I understood correctly, authors use the locations of the observed individuals to define the population locations in their analyses, and extract the stepping stones from an existing habitat suitability map. I’m not sure how they use the preferred land use categories on top of that map. I think this explanation needs clarification. Also, it seems to me that this is related only to stepping stone selection, in which case I suggest merging both sections into one to make the text easier to follow.

Results:
- Line 291. I think in this sentence you mean only northwards?
- Lines 304-306. The acronyms of the provinces might be confusing for an international reader. I suggest to include the full name of the province next to the acronym at least in the Table.
- Lines 307-308. Actually, this sentence is not entirely correct. If you look at the top ten values in your table S3, there is one stepping stone with the same value of a subpopulation. It is only by chance that the subpopulation appears in the arbitrary top ten and not the stepping stone. Please, acknowledge this in your text.
- Lines 354-355. This sentence seems to contradict the previous text. Do you mean that the western populations were still not connected to the big central population?

Discussion:
I appreciate the discussion. My only concern with it is that in my opinion is a bit restrictive. I think a comparison with other connectivity studies and the importance of stepping stones for population connectivity will support your results and provide a broader context.
- Lines 415-419. I think this part needs to be more explained. I guess authors imply that changes in habitat quality come from vegetation succession processes. If so, the link between the two needs to the clearly explained. Additionally, I think the possibility of habitat destruction, should also be included here.
Do the habitat patches where the extinct populations occurred become stepping stones? That could mitigate the impact in the population somehow. I think authors could include some lines about that possibility and its implications. Or if that is the case, if those were used in the current analysis.
- Nodes importance. Although authors acknowledge the importance of stepping stones may have in population viability, I think this is a key result and deserves to be highlighted even further. From a management point of view, it is easier to justify conservation measures where the species occur, but it is more difficult to implement those in areas where the species is not detected.

Figure 1. In this figure is difficult to see the difference between the black lines for provinces and populations. I suggest to redo it with the underlying map in grey. Also, the black dots for the stepping stones might be larger.
The names of the populations here are in Spanish, while they are translated to English in the main text. Please, keep the same throughout the manuscript.

Table 2. This table shows the results for only one of the dispersal distances. Please, include the three distances, as you do with table 3.

Reviewer 2 ·

Basic reporting

1. The manuscript should be thoroughly revised and corrected. I am not a native English speaker, but I feel that the manuscript could improve quite a lot with an additional (also linguistic) revision to improve its readability – even though the spelling is correct throughout the manuscript.
2. The provided literature for the introduction and background is sufficient to see how the manuscript fits into the topic. However, the reference section should be carefully revised since at least one reference is missing (line 499: Rojas et al., 2016).
3. The manuscript, the figures and the tables conform to an acceptable format of “standard sections” according to the provided instructions for PeerJ. The raw data is shared and available (including GIS maps).
4. The manuscript represents an appropriate “unit of publication”. The outline is generally clear, and the study subject is especially relevant for conservation management. Hence it sounds interesting to me.

Experimental design

1. The manuscript topic is within the Aims and Scope of PeerJ.
2. In the present manuscript, the authors did a connectivity modeling exercise with a large set of Dupont’s lark presence data coupled with potential stepping stones based on a MaxEnt probability of presence model. They explore the metapopulation structure (populations and subpopulations) of the endangered Dupont’s lark in Spain and identify critical nodes for the connectivity network. Also, they evaluate different connectivity scenarios according to potential dispersal capacity and the presence of stepping stones in the network.
3. The manuscript deals with a well-defined subject. The objectives are clearly stated at the end of the introduction (lines 130-139). Hypotheses and predictions lack in the introduction. The specific aims of this article are to i) update the cartography of populations and subpopulations of the European Dupont’s lark range, ii) identify both vulnerable and critical nuclei from the connectivity point of view for the conservation of the metapopulation, iii) assess the role of unoccupied but adequate regions in the functionality of the whole metapopulation, iv) testing the effect of different dispersal distances evaluate the degree of isolation of each nucleus, and v) propose adequate conservation measures for the maintenance of the metapopulation.
4. The investigation in this manuscript has been conducted rigorously and to a high technical standard. The research is conforming to the prevailing ethical standards as provided on lines 142-145.
5. The applied methods are sufficiently described to be reproducible by another investigator.

Validity of the findings

1. First of all, I would like to acknowledge the effort made by the authors to transfer scientific knowledge to a management level. The outcomes of this work are of high relevance for the regional authorities in charge of biodiversity management as well as all stakeholders involved in Dupont’s lark conservation and promotion. Hence, I strongly support the publication of such work.
2. All data on which the conclusions are based are provided. The data seems robust, sound and controlled to me.
3. Some additional work on the conclusions (and other parts of the manuscript, see below) is required in my opinion. The authors should carefully revise the vocabulary involving e.g. metapopulation theory and dispersal. For example, a number of potentially identical terms are used, thus confusing at least me (subpopulation, patch, unit, nucleus).
4. As stated in the review guidelines: speculation is welcome, but should be identified as such. I suggest check throughout the discussion, in particular lines 377-379 or lines 492-501.

Additional comments

1. The title “Medium-distance dispersal and stepping stones keep connectivity in Spanish metapopulation of Dupont’s lark.” (lines 1-3) is not precise enough in my opinion. I suggest: “Connectivity in Spanish metapopulation of Dupont’s lark may be maintained by dispersal over medium-distance range and stepping stones.” or similar.

2. Be careful with your use of “dispersal” (lines 97-102, 109-120, throughout the discussion). Dispersal distances are usually described by a dispersal kernel that gives the probability distribution of the distances covered by a number of individuals of a given species, and not just one distance. Hence, you should (throughout the manuscript) be more carefully when talking about your applied criteria, which is namely a dispersal distance threshold (distance beyond which dispersal does not occur).

3. Again (lines 451-458), use “dispersal” carefully. Dispersal is an ecological process that involves the movement of an individual or multiple individuals away from the population in which they were born to another location, or population, where they will settle and reproduce. I would therefore avoid mixing dispersal and movements to avoid adverse conditions (i.e. snow, storm etc.), cf "vagrants" in Supplemental Data S2!

4. A crucial topic (in my opinion) is, however, not discussed in the article (e.g. lines 303-309). The connectivity model only includes information about the habitat surface, the habitat quality and the number of patches of each subpopulation and stepping stone (hence considering the presence and absence of the study species). However, the population size of the subpopulations (i.e. number of territorial males of the Dupont’s lark) is never accounted for in the connectivity model. To give an example: the subpopulation 12 “Pinilla del Campo” has a very high AVS value of 1090.66 (HS 4742, HQ 0.23, n° of patches 1), and is ranked as 6th most important node for intra-patch connectivity (dPCintra) with/without stepping stones. Nevertheless, it holds only 3 territorial males! Opposed to this example is the subpopulation 14 “Aranda de Moncayo” with an approximately 5 times lower AVS vale of only 203.39 (HS 4421, HQ 0.46, n° of patches 10), but holding more than 10 times more territorial males (34). This discrepancy should at least be discussed. The authors might as well reconsider the validity of their chosen approach if such differences between AVS values and the population size of subpopulations are frequent. An additional issue: I feel that the number of patches has a strong (and maybe unwanted) influence on the AHS value. How did you define a patch? How did you determinate the number of patches? Did you analyze the relation between n° of occupied territories and AHS?

5. I think that the authors should discuss the values in Table 1 with regard to the different potential importance of dispersal threshold vs. considering stepping stones. The Equivalent Connectivity Index (EC) comparison in Table 1 shows following “ranking”: EC 5km dispersal threshold/without stepping stones (lowest value) < EC 20km dispersal threshold/without stepping stones < EC 5km dispersal threshold/with stepping stones < EC 100km dispersal threshold/without stepping stones < EC 20km dispersal threshold/with stepping stones < EC 100km dispersal threshold/with stepping stones (highest value)

Annotated reviews are not available for download in order to protect the identity of reviewers who chose to remain anonymous.

---

## Round 0.2 · Minor Revisions

· Academic Editor

Minor Revisions

While I appreciate that you revised the manuscript following most suggestions of the reviewers, and this version is considerably improved, you seem to have neglected responding to my comments in my original annotated pdf. Thus, I must ask that you go back to that annotated manuscript AND this one I am sending to improve clarity and quality of writing, AND address those comments in your next rebuttal letter.

Note that your numbers should be consistent. I would suggest that you recognize that fractions of distance (km) are seldom important, and you round to the nearest kilometer. Also, that fractions of percentages are unimportant and you round to whole numbers. Please pay close attention to general format requirements, some of which were noted by the reviewer.

Note that you have a particular tendency to abuse the term "according to" and which I mentioned the first time. I mention it again this time for you to pay particular attention. You will find many examples of how I improved the clarity of your sentences - feel free to recognize the patterns elsewhere and improve them as necessary.

Reviewer 2 ·

Basic reporting

no comment

Experimental design

no comment

Validity of the findings

no comment

Additional comments

I am satisfied with the revised manuscript, which has clearly improved compared to the previous version – also due to the suggestions made by the second reviewer.
Almost all points made in my first review have been considered. I appreciate your effort to improve the readability of the text by correcting, reformulating, and shortening a series of sentences in the manuscript. You took in account my concern raised about the use of “dispersal” and your consequent change to “movement”. Although you unfortunately have chosen to not discuss the discrepancies between the AVS values and the population size of subpopulations, I am satisfied with your additional explanations regarding the AHS concept.
You have also resolved all other (minor) issues I had pointed out. I found just a few (mostly) spelling or format errors in the manuscript as noted in the attached pdf file.

Annotated reviews are not available for download in order to protect the identity of reviewers who chose to remain anonymous.

---

## Round 0.3 · Minor Revisions

· Academic Editor

Minor Revisions

This newest version is considerably improved and we appreciate your efforts. I have a few minor considerations that I detail in my annotated manuscript. Mostly, some minor wording, some clarification, and some corrections are needed. Please note that you are inconsistent in using commas (,) and periods (.) to indicate decimals. This happens in both the figures and the tables, and perhaps in the text, so please ensure that you standardize all decimal places with periods. Please find the rest of my comments in the annotated manuscript.

---

## Round 0.4 · accepted · Accept

· Academic Editor

Accept

Persistence pays off! Thank you for your efforts in revising the manuscript. I find that the study is much more readable and coherent now after these efforts.